# The Effect of CKD-495, Eupacidin, and Their Marker Compounds on Altered Permeability in a Postoperative Ileus Animal Model

**DOI:** 10.3390/medicina60101707

**Published:** 2024-10-18

**Authors:** Min-Jae Kim, Zahid Hussain, Young Ju Lee, Hyojin Park

**Affiliations:** Department of Internal Medicine, Gangnam Severance Hospital, Yonsei University College of Medicine, Seoul 06274, Republic of Korea; kmj0630@yuhs.ac (M.-J.K.);

**Keywords:** ileus, postoperative complications, permeability, natural products

## Abstract

*Background and Objectives:* Postoperative ileus (POI) is a delay in gastrointestinal transit following surgery that leads to various complications. There is limited understanding of its effective treatment options. CKD-495 and eupacidin are natural products licensed for treating mucosal lesions in acute and chronic gastritis; however, little is known about their effects on intestinal permeability. This study evaluated the effects of CKD-495, eupacidin, and its components (eupatilin and cinnamic acid) on intestinal permeability in an animal model of POI. *Materials and Methods:* Guinea pigs underwent surgical procedures and were randomly assigned to different treatment groups. Drugs were administered orally prior to surgery. Intestinal permeability, leukocyte count, and the expression of calprotectin and tight junction proteins were measured in the harvested ileum tissue. *Results:* The intestinal permeability and leukocyte count were higher in the POI group than in the control group. The pre-administration of CKD-495, cinnamic acid, eupacidin, and eupatilin effectively prevented these changes in the POI model. No significant differences were observed in the expression of tight junction proteins. *Conclusions:* CKD-495, cinnamic acid, eupacidin, and eupatilin exerted protective effects against increased intestinal permeability and inflammation in an animal model of POI. These natural products have potential as therapeutic options for the treatment of POI.

## 1. Introduction

Postoperative ileus (POI) is a common and debilitating complication resulting from delayed gastrointestinal transit following surgery, often leading to increased patient discomfort, prolonged hospital stays, and significant healthcare costs [1,2,3,4]. Although POI can occur after any type of surgery, it is particularly associated with abdominal procedures, where it remains a major unresolved clinical problem [5].

Despite the widespread occurrence of POI, effective therapeutic interventions are lacking, largely due to an incomplete understanding of its pathophysiology [6]. Numerous studies have investigated the underlying mechanisms of POI, revealing the involvement of inflammatory, pharmacological, hormonal, and neurogenic factors [7]. In a previous study on an animal model of POI, increased intestinal inflammation and permeability seemed to be key factors of POI occurrence [8,9].

Natural anti-inflammatory products are an attractive and safe alternative for modulating inflammatory disorders. The herb Artemisia asiatica Nakai is a traditional oriental medicine that has been used for the treatment of diseases such as inflammation, microbial infection, and cancer [10]. Eupacidin, an Artemisia herb isopropanol soft extract, is an orally administered drug manufactured by Chong Kun Dang Pharmaceutical. In Korea, it is indicated to improve gastric mucosal lesions (erosion, bleeding, redness, and edema) in acute and chronic gastritis. The drug contains eupatilin as the active ingredient, which corresponds with 30 mg of the active ingredient per 2000 mg.

Eupatilin (5,7-dihydroxy-3′,4′,6-trimethoxyflavone), a pharmacologically active flavonoid mainly found in the genus Artemisia, is known to possess anti-cancer, anti-inflammatory, antioxidant, neuroprotective, anti-allergic, and cardioprotective activities [11]. Several studies on eupatilin reported that it has the capacity for a variety of biological activities, including a protective effect against NSAID-induced enteropathy [10].

CKD-495, developed by Chong Kun Dang Pharmaceutical, a component extracted from Cinnamomum cassia, shows an effective protective effect compared to Artemisiae Argyi Folium extract (eupacidin, eupatilin, etc.) against acute or chronic gastritis in a phase III trial [12].

Cinnamomum verum and C. cassia Blume are collectively called Cortex Cinnamonmi because of their medicinal cinnamon bark. Cinnamomum verum is more popular elsewhere in the world, whereas C. cassia is widely used in traditional Chinese medicine [13]. Cinnamic acid is an organic acid that occurs naturally in cinnamon bark and has low toxicity and a broad spectrum of biological activity, including an antioxidant effect, free radical scavenging properties, antimicrobial activity, efficacy for diabetes, neurological disorders, and cancer [14,15,16].

We hypothesized that eupacidin and CKD-495 have been shown to exert anti-inflammatory effects by modulating NF-κB signaling and reducing the production of pro-inflammatory cytokines, potentially mitigating the adverse effects of POI on intestinal permeability. In this study, the prevention and treatment of POI was clinically evaluated by confirming the effect of oral eupacidin, CKD-495, and its indicator components (eupatilin and cinnamic acid) on intestinal permeability in an animal model of POI. Our aim is to provide an experimental basis for and predict the effect of drug repositioning CKD-495.

## 2. Materials and Methods

### 2.1. Preparation of Animals

Adult male Hartley guinea pigs (Orient Bio Inc., Seongnam-si, Republic of Korea), weighing 300–350 g, were used in the present study. They were acclimated to controlled breeding conditions (21 ± 1 °C, 50 ± 10% humidity, and 12 h light/dark cycle commencing at 7 a.m.), and a standard guinea pig diet and water were provided for at least one week prior to surgery. All experimental procedures were reviewed and approved by the Institutional Animal Care and Use Committee, Department of Laboratory Animal Resources, Yonsei Biomedical Research Institute, Yonsei University College of Medicine, with Institutional Review Board (IRB) protocol number 2019-0156.

### 2.2. Group Setting

Guinea pigs were randomly assigned to different experimental groups using a computer-generated randomization sequence. Each animal was assigned a group number prior to the commencement of the experiment, and the researchers involved in treatment administration and data collection were blinded to the group assignments to minimize bias. Each group was subjected to a surgical procedure; after maintaining a fasting state (except water) for 24 h before the procedure, a mixture of Zoletil (4 mL), Rumpun (2 mL), and Saline (8 mL) was injected into the abdominal cavity to induce unconsciousness. After 15 min, the hair on the abdomen was removed and the abdominal skin was disinfected with an alcohol swab. A minimal peritoneal incision was made after incising the skin and muscle layers of the abdomen. After abdominal incision, the group with sutures without intestinal manipulation was used as a control group. Other groups underwent cecum extraction, were gently rubbed with wet gauze for 1 min using fingers, and were then sutured. The drug test groups were orally administered the following drugs every 12 h from day 0 to day 2 before surgery: eupacidin = 2000 mg/kg; eupatilin = 30 mg/kg; CKD-495 = 1250, 1670, and 2500 mg/kg; and cinnamic acid = 16.3, 21.7, and 32.5 mg/kg. The drug dosages were determined based on previous animal studies. The animal dose of eupatilin was set at 30 mg/kg based on a previous study by another research team using DA-6034, [17] a similarly active derivative of Artemisia extracts, and eupacidin was set at 2000 mg/kg, which is equivalent to 30 mg/kg of eupatilin. The doses of CKD-495 were set at 1670 mg/kg and 2500 mg/kg to match the 2000 mg/kg dose of eupacidin, based on the clinical dose trial that tried 50 mg and 75 mg of CKD-495 compared to 60 mg of eupacidin, respectively. We added a 1250 mg/kg group, which is half the dose of 2500 mg/kg. Cinnamic acid, which was selected as an indicator component of CKD-495, was administered at 16.3 mg, 21.7 mg, and 32.5 mg/kg, considering that it contains 13 mg per gram of CKD-495.

The number of guinea pigs in each group was 5–8. While tissue samples were obtained from the ileum of each guinea pig used in this study, POI and drug group tissue samples were specifically harvested 3 h after surgery. The 3 h post-surgical time point was chosen based on previous studies indicating that intestinal inflammation and permeability increases significantly within this period following surgical manipulation in guinea pig [8,9]. We used CO_2_ gas to euthanize the guinea pigs before harvesting their intestinal tissues.

Various markers were measured in the harvested ileum. An Ussing chamber (EM-CSYS-2; Physiologic Instruments, San Diego, CA, USA) was used to analyze intestinal permeability, while leukocyte count and calprotectin expression were measured to evaluate intestinal inflammation. We measured the expression of claudin-1 and -2 to evaluate the alterations in the tight junction proteins. We compared the measurements between the control and POI groups and between the POI and drug groups.

### 2.3. Intestinal Permeability

To evaluate intestinal permeability, the harvested tissues were placed in a modified Ussing chamber. Krebs–Ringer bicarbonate solution (2 mL) was added to each half chamber, and the mucosal and serosal sides of the specimens were bathed. At a temperature of 37 °C, both sides were maintained in a gas mixture of 95% O_2_ and 5% CO_2_. After an equilibration period of 30 min, the KRB containing horseradish peroxidase (HRP) at a final concentration of 0.4 mg/mL was substituted for the KRB in the chamber of the mucosal side. The KRB on the serosal side was replaced with fresh KRB: a 0.3 mL sample was collected and replaced with 0.3 mL KRB. Samples from the chamber on the serosal side were enzymatically analyzed using the modified Worthington method. o-Dianisidine dihydrochloride (OPD; Sigma Chemical Co., St Louis, MO, USA) was used as the substrate. Samples (50 μL) were transferred to a microtiter, and 100 μL of an OPD working solution (stable peroxide buffer diluted 1:10 in OPD) was added to each well. Subsequently, the plates were incubated at room temperature while shaking at 300 rpm. After 30 min, 2.5 M sulfuric acid (100 μL) was added. After 10 min, the absorbance of the decolorized reaction product was measured at 492 nm using a microplate reader (Model 680; Bio-Rad Laboratories Inc., Hercules, CA, USA). All samples were analyzed in duplicate and measured using a standard curve. The HRP flux is represented as ng/2 h/mm^2^ during steady-state permeation. Intestinal permeability via the Ussing chamber was expressed as a percent change compared to the mean value of the control group.

### 2.4. Intestinal Inflammation

Histological sections were obtained from the muscle layers of the harvested ileum and proximal colon. The sections were fixed in 10% neutral-buffered formalin and embedded in paraffin. Embedded sections were sliced into 4 μm thickness and then subjected to hematoxylin and eosin staining. We compared the number of leukocytes per high-power field between the control and POI groups and between the vehicle and probiotic groups.

### 2.5. Tight Junction Proteins

The expression of claudin-1 and -2 was determined using immunofluorescence analysis. At 3 h after the operation, the histological sections from the ileum were fixed in 4% paraformaldehyde, embedded with paraffin, and sliced into 4 μm thick sections. Tissues were then deparaffinized, rehydrated, and rinsed using standard methods. The slide sections were incubated overnight with the primary antibody for claudin-1 (1:50; Invitrogen, Carlsbad, CA, USA) or claudin-2 (1:200; Invitrogen) at 4 °C, followed by washing and incubation with the secondary antibody goat anti-rabbit IgG-fluorescein isothiocyanate (1:200; Santa Cruz Biotechnology) for 30 min at 37 °C. The stained samples were evaluated under a fluorescence microscope (Zeiss Axio Imager Z1; Carl Zeiss, Jena, Germany) and the images were analyzed using MetaMorph microscopy automation (MDS Analytical Technologies, Sunnyvale, CA, USA) and ImageJ (NIH and LOCI, University of Wisconsin, Madison, WI, USA).

### 2.6. Statistical Method

The data are expressed as the mean ± SE, and statistical analysis was performed by a non-parametric test using the Mann–Whitney U test between the two groups and a Kruskal–Wallis H test for multiple comparisons. Non-parametric tests were employed due to the small sample sizes in each group (5–8 animals per group), which may result in deviations from normal distribution. SPSS version 26.0 (IBM Corp, Armonk, NY, USA) was used for statistical analysis. A two-tailed *p*-value of < 0.05 indicated statistical significance.

## 3. Results

### 3.1. Intestinal Permeability

The intestinal permeability was indirectly confirmed through a horseradish peroxidase (HRP) absorbance measurement in six subjects in the normal control group, eight in the POI group, six in the eupacidin 2000 mg/kg administration group, seven in the eupatilin 30 mg/kg administration group, six in the CKD-495 doses (1250, 1670, 2500 mg/kg) administration groups, five in the cinnamic acid 16.3 mg/kg administration group, six in the cinnamic acid 21.7 mg/kg administration group, and five subjects in the cinnamic acid 32.5 mg/kg ad-ministration group. HRP absorbance significantly increased in the POI group compared with the control group (0.07 ± 0.03 vs. 0.36 ± 0.4, *p* = 0.01) and decreased in all drug groups compared with the POI group (Figure 1).

### 3.2. Intestinal Inflammation

The intestinal inflammatory grade was measured using the leukocyte count of the same number of subjects in the permeability study, six subjects in the normal control group, eight in the POI group, six in the eupacidin group, seven in the eupatilin group, six in the CKD-495 doses (1250, 1670, 2500 mg/kg) groups, and five/six/five in the cinnamic acid groups (16.3 mg/kg, 21.7 mg/kg 32.5 mg/kg). Leukocyte counts significantly increased in the POI group compared with the control group (25.95 ± 9.92 vs. 38.23 ± 14.15, *p* = 0.004) and decreased in all drug groups compared with the POI group (Figure 2).

### 3.3. Tight Junction Proteins

Expression levels of the tight junction proteins claudin-1 and claudin-2 were measured by analyzing immunofluorescence-stained tissue slide images of the ileum (Figure 3 and Figure 4). We also used a semi-quantitative approach to analyze the slide images using a program to quantify the degree of expression. The results of each group were expressed as the percent change from the mean of the control group, but no significant differences were found between the groups (Figure 5 and Figure 6). (*Claudin-1 and claudin-2 staining in the CKD-495 1675 mg/kg group and claudin-2 staining in the eupatilin group could not be performed due to sample issues.)

## 4. Discussion

This study demonstrates that pre-administration of CKD-495, eupacidin, eupatilin, and cinnamic acid significantly reduces intestinal permeability and leukocyte infiltration in a guinea pig model of postoperative ileus (POI). These findings suggest that these compounds may serve as effective prophylactic treatments for POI, potentially improving postoperative outcomes.

Both eupatilin and eupacidin were effective in preventing increases in intestinal permeability and inflammation in the POI animal model. Eupatilin, the active component of eupacidin, has been previously demonstrated to exert potent anti-inflammatory effects in various models. Notably, a prior study conducted in rats showed that eupatilin dose-dependently suppressed the lipopolysaccharide (LPS)-induced expression of inducible nitric oxide synthase (iNOS) and the subsequent production of nitric oxide (NO), a key mediator in inflammatory responses [18]. This suppression was accompanied by a decrease in nuclear factor (NF)-κB-dependent inflammatory mediators and pro-inflammatory cytokines, including cyclooxygenase-2 (COX-2), monocyte chemoattractant protein-1 (MCP-1), tumor necrosis factor-α (TNF-α), interleukin (IL)-1β, and IL-6 [19]. Given that eupacidin contains eupatilin as its principal active ingredient, it is not surprising that the effects of these two treatments were similar in our study. Both eupatilin and eupacidin significantly inhibited the increases in intestinal permeability and inflammation induced by POI, as evidenced by reduced leukocyte infiltration. The comparison between the eupatilin and eupacidin treatment groups revealed no statistically significant differences in their efficacy, suggesting that the therapeutic effects of eupacidin in this model can be largely attributed to its eupatilin content. These findings align with previous studies and further support the potential of eupatilin as a key anti-inflammatory agent in the management of POI.

CKD-495, alongside its key active component, cinnamic acid, effectively inhibited POI-induced increases in intestinal permeability and inflammation across all tested doses in an animal model. The similar mean values observed between the CKD-495 and cinnamic acid treatment groups suggest that cinnamic acid is the primary contributor to these protective effects. CKD-495 was recently recognized for its safety and efficacy in a Phase III clinical trial, where it demonstrated a significant ability to improve gastric mucosal lesions in patients with acute and chronic gastritis [12]. Following its successful trial outcomes, CKD-495 has been released as a commercial product, expanding its potential applications. Cinnamic acid is a well-known organic acid found in natural plants, characterized by its low toxicity and broad spectrum of biological activities, particularly its anti-inflammatory effects. Previous studies have reported that its antioxidant activity, achieved through direct scavenging of reactive oxygen species and inhibition of lipoxygenase, contributes to its anti-inflammatory properties [20,21]. Additionally, a recent study has shown that cinnamic acid can regulate the intestinal microbiome and short-chain fatty acids, which are closely associated with intestinal inflammation and permeability [22]. We believe that these specific mechanisms underlie the observed prevention of intestinal inflammation and the reduction in permeability in this study. Despite varying the doses of CKD-495 and cinnamic acid, no significant differences in intestinal permeability were detected between the different dosage groups, indicating a lack of dose-dependency within the tested range. Increasing the dosage did not result in further reductions in permeability beyond what was observed in the normal control group, implying that the therapeutic ceiling for these compounds might have been reached at the lower doses. This finding is crucial for informing dosing strategies in future clinical studies, as it suggests that lower doses of CKD-495 and cinnamic acid may be adequate for achieving the desired therapeutic effects, potentially minimizing the risk of side effects associated with higher doses. Future research should focus on validating these results in human studies and exploring the long-term efficacy and safety of these dosing strategies. Additionally, exploring the pharmacokinetics and pharmacodynamics of CKD-495 and cinnamic acid could provide deeper insights into optimizing their clinical use.

Previous studies by our group have demonstrated that the increased permeability of the intestinal wall following surgical manipulation is accompanied by intestinal wall inflammation [8,9]. Therefore, it is reasonable to conclude that modulating this increased permeability would be beneficial in preventing the development of POI. In this study, eupacidin, eupatilin, CKD-495, and cinnamic acid were all effective in preventing the increase in intestinal permeability and inflammation in the POI guinea pig model when administered preoperatively. Thus, all four agents used in this study have the potential to inhibit the development of POI if administered preoperatively.

We measured the expression of the representative tight junction (TJ) proteins, claudin-1 and claudin-2, which are critical in maintaining the intestinal barrier while regulating the permeability of ions, water, and nutrients [23]. Since previous studies have suggested that changes in the expression of claudin-1, claudin-2, and other TJ proteins may affect the development of POI, we hypothesized that similar changes would occur in our POI group compared to the control group and that the pre-administration of the drug would inhibit the development of POI [9]. However, no significant differences in the TJ protein expression were observed between the control and POI groups in this experiment, nor among the drug treatment groups. The three-hour time point was selected based on previous animal studies showing that the ileal contraction decreases most significantly three hours after surgery [8]. However, this time point may not be optimal for detecting changes in the TJ protein expression related to POI development. In addition to the timing of tissue collection, several other factors may have contributed to the lack of significant differences in the tight junction protein expression observed in this study. Tight junction protein dynamics are known to be highly responsive to various stimuli, and it is possible that changes in claudin-1 and claudin-2 occur at time points beyond the three-hour window. Moreover, the specific tight junction proteins evaluated (claudin-1 and claudin-2) may not fully reflect the broader regulatory mechanisms involved in intestinal barrier function. Future research should consider evaluating other proteins such as occludin and ZO-1, which may provide more comprehensive insights into the regulation of tight junctions during POI.

This study has several limitations. First, the small sample size in each experimental group (5–8 animals) may have reduced the statistical power to detect more subtle effects on the tight junction protein expression and inflammatory markers. A larger sample size could provide more robust conclusions regarding the efficacy of CKD-495 and cinnamic acid. Second, the dosing regimens used in this study were based on previous preclinical trials; however, the lack of a clear dose–response relationship suggests that further investigation is required to optimize the therapeutic doses of these compounds. Future studies should focus on testing a wider range of doses and employing larger sample sizes to determine the most effective therapeutic window and to confirm these findings in clinical settings. Lastly, the timing of tissue collection (3 h post-surgery) may not have been sufficient to capture dynamic changes in the tight junction protein expression, as other studies have observed significant effects at later time points. Extending the observation period in future studies may provide additional insights into the role of these compounds in modulating intestinal barrier function.

## 5. Conclusions

In conclusion, increased intestinal permeability and inflammation are critical factors in the pathogenesis of POI. This study demonstrates that eupacidin, eupatilin, CKD-495, and cinnamic acid have protective effects against these pathological changes. Therefore, these four compounds represent promising therapeutic agents for the prevention and treatment of POI.

## Figures and Tables

**Figure 1 medicina-60-01707-f001:**
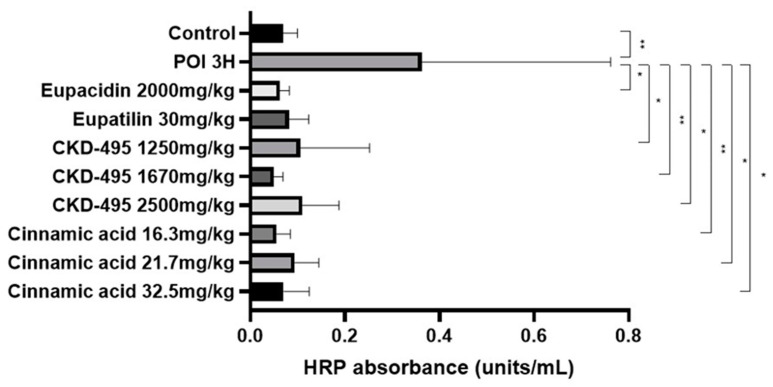
Horseradish peroxidase (HRP) absorbance results (units/mL) for each guinea pig group, measured 3 h after surgery to assess intestinal permeability. The POI group exhibited a significant increase in HRP absorbance compared to the control group, while all treatment groups showed significant reductions in HRP absorbance, indicating decreased permeability. Bars represent the mean ± SEM. (* *p* < 0.05, ** *p* < 0.01 compared to the control group).

**Figure 2 medicina-60-01707-f002:**
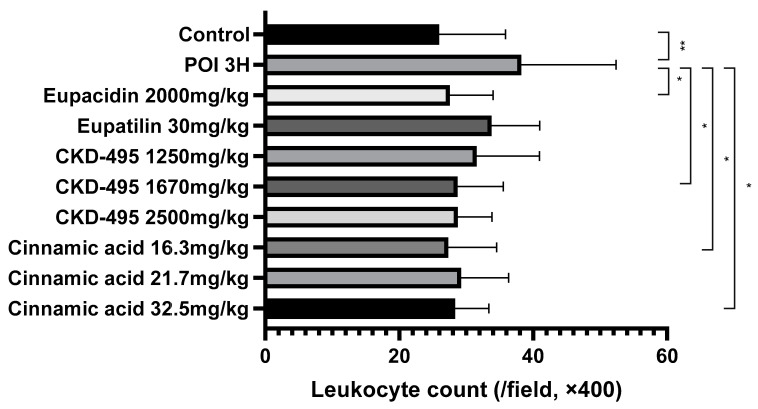
Leukocyte counts per high-power field (400× magnification) for each guinea pig group, 3 h after surgery. The POI group showed significantly increased leukocyte infiltration compared to the control group, while all treatment groups (CKD-495, eupatilin, eupacidin, and cinnamic acid) exhibited significantly reduced leukocyte counts, indicating anti-inflammatory effects. Bars represent the mean ± SEM. (* *p* < 0.05, ** *p* < 0.01 compared to the control group).

**Figure 3 medicina-60-01707-f003:**
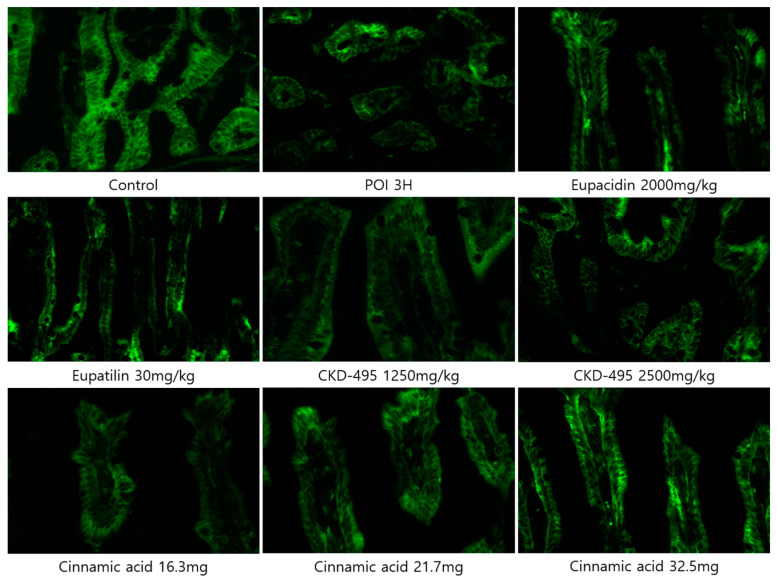
Immunofluorescence-stained images showing claudin-1 expression in ileal tissue sections from guinea pigs in each experimental group.

**Figure 4 medicina-60-01707-f004:**
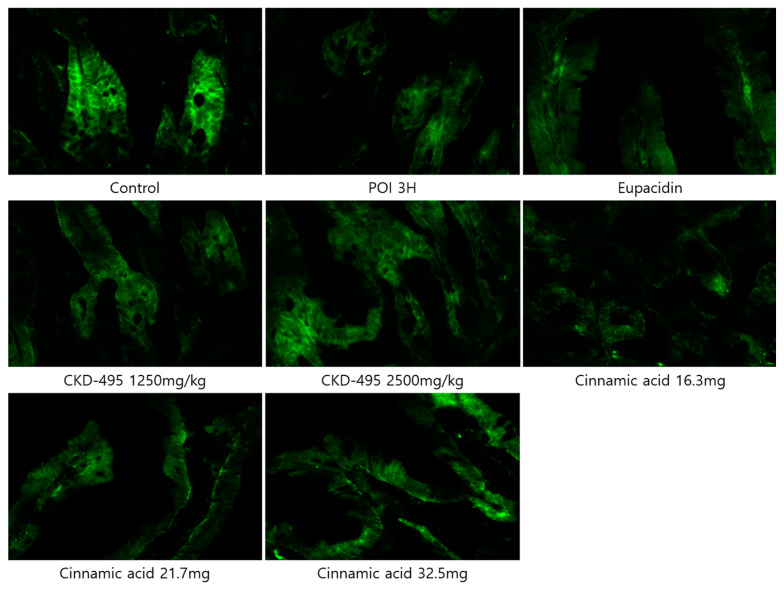
Immunofluorescence-stained images showing claudin-2 expression in ileal tissue sections from guinea pigs in each experimental group.

**Figure 5 medicina-60-01707-f005:**
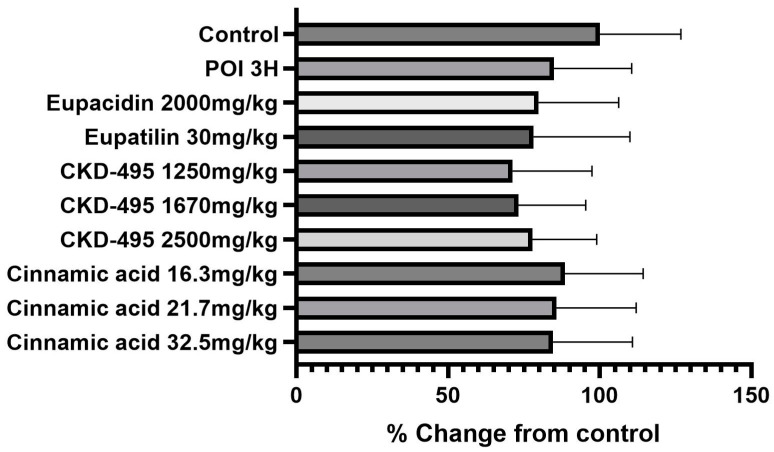
Quantification of claudin-1 expression levels in each guinea pig group. Bars represent the mean ± SEM. No significant differences were observed between the experimental groups.

**Figure 6 medicina-60-01707-f006:**
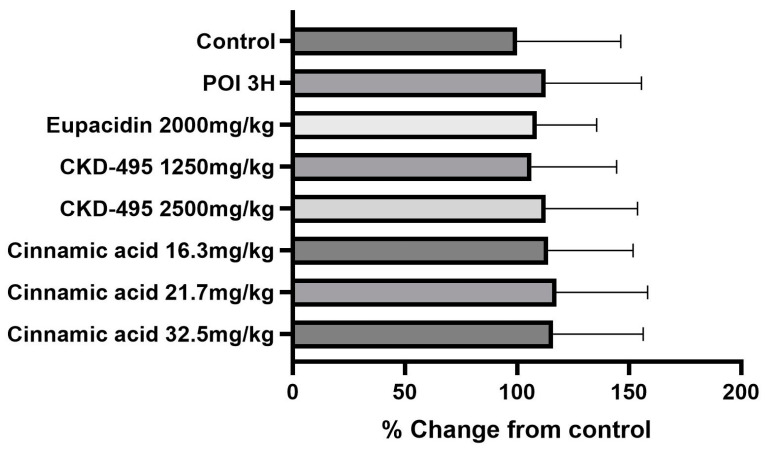
Quantification of claudin-2 expression levels in each guinea pig group. Bars represent the mean ± SEM. No significant differences were observed between the experimental groups.

## Data Availability

The original contributions presented in the study are included in the article, further inquiries can be directed to the corresponding author.

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
