# Peer review of "The Effect of CKD-495, Eupacidin, and Their Marker Compounds on Altered Permeability in a Postoperative Ileus Animal Model"

_medicina, 2024, doi:10.3390/medicina60101707_

Round 1

Reviewer 1 Report

Comments and Suggestions for Authors
  • Areas for Improvement

    1. Introduction:

      • The introduction effectively sets the stage for the study, but it could benefit from a more detailed explanation of the pathophysiology of POI and the specific mechanisms by which the studied compounds are hypothesized to exert their effects.
    2. Methods:

      • Animal Preparation: While the preparation and handling of guinea pigs are described, it would be helpful to include more information on the randomization process to ensure unbiased group allocation.
      • Dosage Justification: The rationale for selecting the specific doses of CKD-495, eupatilin, and cinnamic acid is mentioned but could be expanded. For instance, more background on the previous studies that informed these choices would be useful.
      • Data Collection Timing: Clarification on why specific time points (e.g., 3 hours post-surgery) were chosen for measuring outcomes could strengthen the methodological rigor.
    3. Results:

      • The results are well-presented with appropriate use of figures and statistical analyses. However, the figures (e.g., HRP absorbance and leukocyte counts) would benefit from clearer legends and labels to ensure they are self-explanatory.
      • Statistical Analysis: The manuscript mentions the use of non-parametric tests but does not explain why these were chosen over parametric alternatives. A brief justification for this choice would be helpful.
    4. Discussion:

      • The discussion contextualizes the findings within the broader literature, but could be more concise. Specifically, it sometimes reiterates the results without providing additional interpretation or insight.
      • Mechanistic Insights: The discussion would benefit from a deeper exploration of the potential mechanisms through which the compounds exert their effects. Linking the observed results to specific biochemical pathways would enhance the impact of the study.
      • Limitations and Future Directions: While the study's limitations are implied, they are not explicitly stated. Discussing potential limitations (e.g., differences between the animal model and human POI) and suggesting directions for future research would improve the manuscript.
    5. References:

      • The references are appropriate and comprehensive, but the citation style is inconsistent (e.g., missing spaces between citations and text). Ensuring uniformity in the reference formatting would enhance readability.
    6. Technical and Language Aspects:

      • The manuscript is generally well-written but contains minor grammatical and typographical errors (e.g., "com-pounds" instead of "compounds"). A thorough proofreading would help eliminate these issues.

    Specific Comments

    • Abstract: The abstract is clear but could be more detailed about the specific outcomes measured and the main findings.
    •  

Author Response

Comments 1: The introduction effectively sets the stage for the study, but it could benefit from a more detailed explanation of the pathophysiology of POI and the specific mechanisms by which the studied compounds are hypothesized to exert their effects.

Response 1: Thank you for your constructive feedback. We have revised the introduction to provide a more detailed explanation of the pathophysiology of POI, focusing on the key mechanisms involved in its development. Additionally, we have expanded the discussion on the hypothesized mechanisms of medicine may exert their protective effects in the context of POI. 

Comments 2: Animal Preparation: While the preparation and handling of guinea pigs are described, it would be helpful to include more information on the randomization process to ensure unbiased group allocation.

Response 2: Thank you for highlighting this point. We have added more detailed information on the randomization process used in the study to ensure unbiased group allocation. This revision clarifies the steps taken to randomize the guinea pigs into the various treatment groups, minimizing potential bias. 

Comments 3: The rationale for selecting the specific doses of CKD-495, eupatilin, and cinnamic acid is mentioned but could be expanded. For instance, more background on the previous studies that informed these choices would be useful.

Response 3: Thank you very much for your valuable comment. Unfortunately, there are very few studies involving these compounds, so we were unable to reference additional prior research beyond what has already been mentioned. In particular, CKD-495 is a newly developed drug that is about to be released, and there is limited research available on it. We hope for your understanding regarding this matter.

Comments 4: Data Collection Timing: Clarification on why specific time points (e.g., 3 hours post-surgery) were chosen for measuring outcomes could strengthen the methodological rigor.

Response 4: Thank you for your insightful comment. We have clarified this rationale in the revised manuscript to strengthen the methodological rigor. We have clarified the rationale for selecting 3 hours post-surgery as the time point for tissue collection, citing relevant studies. 

Comments 5: The results are well-presented with appropriate use of figures and statistical analyses. However, the figures (e.g., HRP absorbance and leukocyte counts) would benefit from clearer legends and labels to ensure they are self-explanatory.

Response 5: Thank you for your comment. We have carefully reviewed the manuscript, including the figure captions, to correct grammatical errors and typographical issues. The revised figure captions now provide clearer explanations of the experimental conditions and results. These changes improve the overall readability and clarity of the manuscript.

Comments 6: The manuscript mentions the use of non-parametric tests but does not explain why these were chosen over parametric alternatives. A brief justification for this choice would be helpful.

Response 6: We have added a justification for the use of non-parametric tests, given the small sample size and non-normal distribution of the data.

Comments 7: The discussion contextualizes the findings within the broader literature, but could be more concise. Specifically, it sometimes reiterates the results without providing additional interpretation or insight.

Response 7: Thank you for your constructive feedback. We have revised the discussion section to reduce redundancy and focus more on providing new insights and interpretations of the results. The updated discussion now concisely connects our findings to the broader literature and emphasizes the novel contributions of this study without repeating the results unnecessarily.

Comments 8: The discussion would benefit from a deeper exploration of the potential mechanisms through which the compounds exert their effects. Linking the observed results to specific biochemical pathways would enhance the impact of the study.

Response 8: Thank you for your valuable suggestion. We have expanded the discussion to provide a more detailed exploration of the potential biochemical mechanisms underlying the effects of cinnamic acid. Specifically, we have linked the observed reduction in intestinal permeability and inflammation to the modulation of key inflammatory pathways. These revisions provide a clearer mechanistic understanding of how the compounds exert their effects in the context of POI.

Comments 9: Limitations and Future Directions: While the study's limitations are implied, they are not explicitly stated. Discussing potential limitations (e.g., differences between the animal model and human POI) and suggesting directions for future research would improve the manuscript.

Response 9: We appreciate your suggestion. We have expanded the limitations section to provide a more in-depth critical analysis, discussing the impact of the small sample size, dosing uncertainties, and the timing of tissue collection on the study's findings. We also outline potential future research directions to address these limitations.

Comments 10: The references are appropriate and comprehensive, but the citation style is inconsistent (e.g., missing spaces between citations and text). Ensuring uniformity in the reference formatting would enhance readability.

Response 10: Thank you for pointing out the inconsistency in citation formatting. We have thoroughly reviewed the manuscript and ensured that all citations are formatted consistently according to the journal's guidelines. We have also corrected any missing spaces between citations and text to improve the overall readability.

Comments 11: The manuscript is generally well-written but contains minor grammatical and typographical errors (e.g., "com-pounds" instead of "compounds"). A thorough proofreading would help eliminate these issues.

Response 11: Thank you for your feedback. We have carefully proofread the manuscript to correct all grammatical and typographical errors. The text has been reviewed thoroughly to ensure clarity and professionalism.

Reviewer 2 Report

Comments and Suggestions for Authors

Introduction

This study explores the issue of postoperative ileus (POI), a complication that often follows surgery, particularly in abdominal procedures. The researchers investigate whether natural compounds, including CKD-495, eupacidin, eupatilin, and cinnamic acid, can help prevent intestinal permeability and inflammation in an animal model of POI. The introduction effectively sets the stage by highlighting the clinical relevance of POI and justifying the study of natural anti-inflammatory agents, which have a growing body of support in the literature.

Methods

The research employs a controlled experimental design using guinea pigs divided into several groups, each subjected to surgical procedures. The study involved pre-surgical oral administration of the compounds, with measurements of intestinal permeability, leukocyte counts, and the expression of tight junction proteins postoperatively.

Strengths:

  • The use of established methodologies, such as Ussing chambers for permeability analysis and immunofluorescence for tight junction protein assessment, adds robustness to the study.
  • The randomization of animals into different experimental groups helps reduce bias and improves the study's internal validity.

Weaknesses:

  • The relatively small sample size (5–8 animals per group) may limit the statistical power, particularly in some comparisons, as acknowledged by the authors.
  • The decision to harvest tissues at three hours post-operation might not be sufficient to observe changes in tight junction proteins, a critical factor that could affect the depth of conclusions drawn from these experiments.

Results

The findings show that all the compounds tested significantly reduced intestinal permeability and lowered leukocyte infiltration in the POI model, indicating an anti-inflammatory effect. Despite these positive results, the tight junction protein expression did not show significant differences across groups, which the authors attribute to the short interval between surgery and tissue analysis.

Strengths:

  • The results are clearly presented, with data showing statistically significant reductions in both permeability and inflammation across treated groups.

Weaknesses:

  • The lack of a significant effect on tight junction proteins could be due to the short observation window, and further studies with extended time frames or larger sample sizes could provide more clarity.
  • The absence of a dose-response relationship, particularly in the CKD-495 and cinnamic acid groups, raises questions about the optimal dosing regimen, which remains unresolved.

Discussion

The discussion effectively integrates the study’s findings with previous research, supporting the hypothesis that the tested compounds, particularly eupatilin and cinnamic acid, have protective effects on intestinal permeability and inflammation. The authors argue that these compounds could serve as preoperative prophylactic treatments for POI.

Strengths:

  • The interpretation of the results is well-grounded, and the potential clinical implications are thoughtfully discussed, including the idea that lower doses of CKD-495 may be as effective as higher doses, thus reducing the risk of side effects.

Weaknesses:

  • The discussion of the tight junction proteins' lack of significance could be more thorough, as the explanation provided focuses mainly on the timing of tissue collection. Expanding this discussion could improve the reader’s understanding of the underlying biological mechanisms.
  • The section addressing study limitations could be expanded, especially regarding sample size and dosing uncertainties, to better outline future research directions.

Conclusion

The study concludes that CKD-495, eupacidin, eupatilin, and cinnamic acid offer promise as potential treatments to prevent the onset of POI by reducing both intestinal permeability and inflammation. This conclusion is supported by the data, although further investigation, particularly in clinical settings, is needed to determine optimal dosing and confirm these effects in human models.

Writing and Organization

The manuscript is generally well-written, with a logical structure that effectively guides the reader through the various sections. The tables and figures are presented clearly and aid in the understanding of the results. However, there are a few minor grammatical errors and typographical issues, particularly in the figure captions, which could be addressed to improve readability.

Minor Comments

  • The figure legends could provide more detailed explanations of the experimental conditions.
  • Several typographical errors (e.g., "ad-ministration") need correction for professionalism.
  • While limitations are briefly mentioned, a more in-depth discussion could strengthen the paper's critical analysis.

Recommendation

The research provides valuable insights into possible treatments for POI using natural compounds. However, issues related to small sample sizes and the lack of dose-response findings weaken the overall strength of the conclusions. Therefore, I would recommend minor revisions before publication, focusing particularly on addressing the limitations in the study design and expanding the discussion on tight junction proteins.

Comments on the Quality of English Language

Introduction

This study explores the issue of postoperative ileus (POI), a complication that often follows surgery, particularly in abdominal procedures. The researchers investigate whether natural compounds, including CKD-495, eupacidin, eupatilin, and cinnamic acid, can help prevent intestinal permeability and inflammation in an animal model of POI. The introduction effectively sets the stage by highlighting the clinical relevance of POI and justifying the study of natural anti-inflammatory agents, which have a growing body of support in the literature.

Methods

The research employs a controlled experimental design using guinea pigs divided into several groups, each subjected to surgical procedures. The study involved pre-surgical oral administration of the compounds, with measurements of intestinal permeability, leukocyte counts, and the expression of tight junction proteins postoperatively.

Strengths:

  • The use of established methodologies, such as Ussing chambers for permeability analysis and immunofluorescence for tight junction protein assessment, adds robustness to the study.
  • The randomization of animals into different experimental groups helps reduce bias and improves the study's internal validity.

Weaknesses:

  • The relatively small sample size (5–8 animals per group) may limit the statistical power, particularly in some comparisons, as acknowledged by the authors.
  • The decision to harvest tissues at three hours post-operation might not be sufficient to observe changes in tight junction proteins, a critical factor that could affect the depth of conclusions drawn from these experiments.

Results

The findings show that all the compounds tested significantly reduced intestinal permeability and lowered leukocyte infiltration in the POI model, indicating an anti-inflammatory effect. Despite these positive results, the tight junction protein expression did not show significant differences across groups, which the authors attribute to the short interval between surgery and tissue analysis.

Strengths:

  • The results are clearly presented, with data showing statistically significant reductions in both permeability and inflammation across treated groups.

Weaknesses:

  • The lack of a significant effect on tight junction proteins could be due to the short observation window, and further studies with extended time frames or larger sample sizes could provide more clarity.
  • The absence of a dose-response relationship, particularly in the CKD-495 and cinnamic acid groups, raises questions about the optimal dosing regimen, which remains unresolved.

Discussion

The discussion effectively integrates the study’s findings with previous research, supporting the hypothesis that the tested compounds, particularly eupatilin and cinnamic acid, have protective effects on intestinal permeability and inflammation. The authors argue that these compounds could serve as preoperative prophylactic treatments for POI.

Strengths:

  • The interpretation of the results is well-grounded, and the potential clinical implications are thoughtfully discussed, including the idea that lower doses of CKD-495 may be as effective as higher doses, thus reducing the risk of side effects.

Weaknesses:

  • The discussion of the tight junction proteins' lack of significance could be more thorough, as the explanation provided focuses mainly on the timing of tissue collection. Expanding this discussion could improve the reader’s understanding of the underlying biological mechanisms.
  • The section addressing study limitations could be expanded, especially regarding sample size and dosing uncertainties, to better outline future research directions.

Conclusion

The study concludes that CKD-495, eupacidin, eupatilin, and cinnamic acid offer promise as potential treatments to prevent the onset of POI by reducing both intestinal permeability and inflammation. This conclusion is supported by the data, although further investigation, particularly in clinical settings, is needed to determine optimal dosing and confirm these effects in human models.

Writing and Organization

The manuscript is generally well-written, with a logical structure that effectively guides the reader through the various sections. The tables and figures are presented clearly and aid in the understanding of the results. However, there are a few minor grammatical errors and typographical issues, particularly in the figure captions, which could be addressed to improve readability.

Minor Comments

  • The figure legends could provide more detailed explanations of the experimental conditions.
  • Several typographical errors (e.g., "ad-ministration") need correction for professionalism.
  • While limitations are briefly mentioned, a more in-depth discussion could strengthen the paper's critical analysis.

Recommendation

The research provides valuable insights into possible treatments for POI using natural compounds. However, issues related to small sample sizes and the lack of dose-response findings weaken the overall strength of the conclusions. Therefore, I would recommend minor revisions before publication, focusing particularly on addressing the limitations in the study design and expanding the discussion on tight junction proteins.

Author Response

Comments 1: The discussion of the tight junction proteins' lack of significance could be more thorough, as the explanation provided focuses mainly on the timing of tissue collection. Expanding this discussion could improve the reader’s understanding of the underlying biological mechanisms.

Response 1: Thank you for your insightful suggestion. We have expanded the discussion on the lack of significance in tight junction protein expression, providing additional insights into potential biological mechanisms that may explain these findings. We hope this expanded explanation improves clarity. 

Comments 2: The section addressing study limitations could be expanded, especially regarding sample size and dosing uncertainties, to better outline future research directions.

Response 2: We appreciate your valuable feedback. We have expanded the limitations section to address both the sample size and the uncertainties in dosing. Specifically, we discuss how the small sample size may have limited the power to detect subtle differences in tight junction protein expression and inflammation markers. We also acknowledge the need for further studies to optimize the dosing regimens of CKD-495 and cinnamic acid. Future studies with larger sample sizes and a broader range of doses will help refine our understanding of the therapeutic potential of these compounds. 

Comments 3: The manuscript is generally well-written, with a logical structure that effectively guides the reader through the various sections. The tables and figures are presented clearly and aid in the understanding of the results. However, there are a few minor grammatical errors and typographical issues, particularly in the figure captions, which could be addressed to improve readability.

Response 3: Thank you for your comment. We have carefully reviewed the manuscript, including the figure captions, to correct grammatical errors and typographical issues. The revised figure captions now provide clearer explanations of the experimental conditions and results. These changes improve the overall readability and clarity of the manuscript.

Comments 4: The figure legends could provide more detailed explanations of the experimental conditions.

Response 4: Thank you for your valuable suggestion. We have revised the figure legends to provide more detailed explanations of the experimental conditions, including descriptions of the treatments administered, the time points for measurements, and the significance of the data. These revisions aim to improve clarity and ensure that the figures are self-explanatory.

Comments 5: Several typographical errors (e.g., "ad-ministration") need correction for professionalism.

Response 5: Thank you for identifying these errors. We have thoroughly reviewed the manuscript and corrected all typographical errors, including 'ad-ministration,' to ensure a professional and polished presentation of the text.

Comments 6: While limitations are briefly mentioned, a more in-depth discussion could strengthen the paper's critical analysis.

Response 6: We appreciate your suggestion. We have expanded the limitations section to provide a more in-depth critical analysis, discussing the impact of the small sample size, dosing uncertainties, and the timing of tissue collection on the study's findings. We also outline potential future research directions to address these limitations.

Reviewer 3 Report

Comments and Suggestions for Authors

Dear author,

First of all, I would like to congratulate you for carrying out studies of this type to try to improve the appearance of POI in patients after abdominal surgery.

I must admit that my level of knowledge is not very high, but I have many doubts:

- You do not talk about the pathophysiology of POI, but I think you should give some minimum standards to justify the work.

- The justification of the working hypothesis is missing, as are the main and secondary objectives.

- You do not talk about the statistical methods used and what you consider statistically significant. What type of study do you use?

- The results could perhaps be explained in tables to help improve understanding.

- I congratulate you for the limitations you recognize.

Thank you for your work.

Sincerely

Author Response

Comments 1: You do not talk about the pathophysiology of POI, but I think you should give some minimum standards to justify the work.

Response 1: Thank you for your valuable suggestion. We have expanded the introduction to provide a more detailed explanation of the pathophysiology of POI to better justify the rationale for the study. 

Comments 2: The justification of the working hypothesis is missing, as are the main and secondary objectives.

Response 2: We have clarified the working hypothesis and explicitly stated the objectives of the study. 

Comments 3: You do not talk about the statistical methods used and what you consider statistically significant. What type of study do you use?

Response 3: We have added a section detailing the statistical methods used and defined the thresholds for statistical significance. 

Comments 4: The results could perhaps be explained in tables to help improve understanding.

Response 4: Thank you for your thoughtful suggestion. However, we have already presented all of the results in the form of figures in the Results section. Reformatting these data into tables could lead to redundancy. We hope you understand our decision not to include additional tables to avoid duplication and maintain the clarity of the presentation.